# RE-EVALUATING RETROSYNTHESIS ALGORITHMS WITH SYNTHESEUS

## ABSTRACT

The planning of how to synthesize molecules, also known as retrosynthesis, has been a growing focus of the machine learning and chemistry communities in recent years. Despite the appearance of steady progress, we argue that imperfect benchmarks and inconsistent comparisons mask systematic shortcomings of existing techniques. To remedy this, we present a benchmarking library called SYNTHESEUS which promotes best practice by default, enabling consistent meaningful evaluation of single-step and multi-step retrosynthesis algorithms. We use SYNTHESEUS to re-evaluate a number of previous retrosynthesis algorithms, and find that the ranking of state-of-the-art models changes when evaluated carefully. We end with guidance for future works in this area.

## 1 INTRODUCTION

Over the past five years, the use of machine learning, and generative models in particular, has led to renewed interest in the automated computational design of novel molecules (Segler et al., 2017; Gómez-Bombarelli et al., 2018; Meyers et al., 2021; Maziarz et al., 2022). Although such approaches can help to discover compounds with the desired property profiles more efficiently, most existing methods do not explicitly account for synthesizability, and therefore often output molecules which are hard to synthesize in a wet-lab (Klebe, 2009). This motivates the development of fast and reliable computer-aided synthesis planning (CASP) algorithms which check for synthesizability by explicitly designing synthesis routes for an input molecule (Strieth-Kalthoff et al., 2020), also known as *retrosynthesis*.

Retrosynthesis works by recursively decomposing a target molecule into increasingly simpler molecules using formally reversed chemical reactions, until a set of purchasable or known building block molecules is found. Starting with the building blocks, the reactions in the forward direction provide a recipe of how to synthesize the target. Most work in the area has focused on studying these two components, single-step retrosynthesis models and multi-step planning algorithms, independently.

In *single-step* retrosynthesis, models are given a molecule and output reactions which produce that molecule in one step (Segler & Waller, 2017; Coley et al., 2017; Liu et al., 2017; Dai et al., 2019; Tetko et al., 2020). Recent work in this area has generally focused on training neural networks to predict reactions extracted from the scientific literature or patents (Lowe, 2012; Zhong et al., 2023). In *multi-step* planning, given a target molecule, a set of purchasable molecules, and a single-step retrosynthesis model, the goal is to produce complete synthesis routes. This is challenging as the search space is extremely large compared to the number of solutions. Recent work in this area has used Monte Carlo Tree Search (MCTS), Reinforcement Learning (RL), or heuristic-guided search algorithms, to selectively explore a tree of possible reactions from the starting molecule (Segler et al., 2018; Coley et al., 2019; Schwaller et al., 2020; Chen et al., 2020; Xie et al., 2022; Tripp et al., 2022; Liu et al., 2023).

In this work, we take a closer look at the commonly used metrics for single and multi-step retrosynthesis. First, it is not clear how metrics used when benchmarking single-step and multi-step in isolation should be interpreted in the context of an end-to-end retrosynthesis pipeline. Second, model comparison and metrics use in prior work has been inconsistent. The goal of this paper is to specify best practices for evaluating retrosynthesis algorithms and perform a rigorous re-evaluation and analysis of prior work. To facilitate this, we developed a python package called SYNTHESEUS which allows researchers to evaluate their approaches in a consistent way. The paper is organized as follows. Section 2 examines how retrosynthesis was evaluated in previous works, and points out shortcomings in this practice. We find several previously reported results to be understated, overstated, or otherwise

not comparable to each other. We then present best practices for the field. In Section 3 we present our python package SYNTHESEUS. It supports consistent evaluation of single-step and multi-step retrosynthesis algorithms, with best practice enforced by default. In Section 4 we use SYNTHESEUS to re-evaluate existing single-step and multi-step methods across many settings in an attempt to "set the record straight". We refrain from making a final recommendation about which models or algorithms are the best, as we argue currently existing metrics are not sufficient. Our results instead serve as a starting point for future research, which we hope SYNTHESEUS to accelerate. In Section 5 we examine how our contributions fit in with prior work, and in Section 6 we provide a roadmap of how SYN-THESEUS can contribute to longer-term improvements in retrosynthesis methods and their evaluation.

## 2 PITFALLS AND BEST PRACTICE FOR RETROSYNTHESIS EVALUATION

Evaluation in retrosynthesis is largely constrained by two realities. First, actually performing synthesis in the lab is costly, time-consuming, and requires significant expertise; it is therefore infeasible for most researchers who work on algorithm development, and should not be a requirement, even though experimental validation is clearly important. Second, because of the division into single-step and multi-step, most works seek to evaluate one part of the retrosynthesis pipeline in isolation rather than holistically, while the key to real-world adoption lies in end-to-end performance. Keeping this in mind, in this section we survey the merits and shortcomings of existing evaluation practices for single-step models and multi-step search algorithms.

### 2.1 SINGLE-STEP MODELS

Single-step retrosynthesis models have several functions in CASP programs: (1) defining which reactions are *feasible* ways of obtaining a given molecule, effectively defining the search environment; and (2) *ranking* or otherwise expressing preference over these reactions, effectively acting as a policy or heuristic to guide the search. Most single-step retrosynthesis models output a list of reactions, and are trained using supervised learning to output reactions which were used in real synthesis routes and furthermore to rank these reactions highly. The most common evaluation strategy is to evaluate the top-$k$ accuracy on a held-out test set, i.e. the fraction of molecules where the reaction which occurred in the dataset is ranked in the first $k$ outputs (Segler & Waller, 2017; Liu et al., 2017; Coley et al., 2017). This parallels evaluation commonly used in computer vision (Deng et al., 2009; Krizhevsky et al., 2017). In this section, we explain how this evaluation metric does not fully measure the utility of single-step models in CASP programs, and how subtle differences in evaluation have distorted the numbers reported in prior works. We suggest best practice for each of these points.

**Pitfall S1: measuring recall rather than precision**    By measuring how often reactions from the dataset occur in the model outputs, top-$k$ accuracy essentially tests the model's ability to *recall* the dataset. Unless $k = 1$ and the top-1 accuracy is nearly 100%, a multi-step search algorithm using this single-step model will almost certainly use reactions not contained in the dataset for planning. If these reactions have low quality or feasibility then routes using them will not be useful. On the other hand, in many cases there are several possible ways to make a particular molecule. Therefore, as previously argued by Schwaller et al. (2020), top-$k$ *precision* of a single-step model (what fraction of the top $k$ reactions are feasible) is arguably equally or more important than recall for multi-step search. Unfortunately, without an expert chemist or a wet-lab, precision is hard to measure. Nonetheless, this suggests that models with a higher top-$k$ accuracy are not necessarily more useful in CASP programs. **Best practice:** Although it is not clear how this can be done, we believe authors should strive to evaluate the precision of their models, at the very least through a visual check of several examples. Some prior works use round-trip accuracy using a forward reaction model (also referred to as back-translation) to measure feasibility of reactions that are not necessarily ground-truth (Schwaller et al., 2019; Chen & Jung, 2021). However, we note the inconsistent use of the term "round-trip accuracy" in prior work: Chen & Jung (2021) compute it in a top-$k$ fashion where *at least one* of the top-$k$ results has to round-trip in order for the prediction to count as successful, which does *not* measure precision; in (Schwaller et al., 2020) this metric is called *coverage*. Round-trip accuracy also relies on a fixed forward model, which is usually only trained on real reactions (i.e. is given sets of reactants that actually react as input) without the presence of negative data; it is unclear whether such a model can be used to evaluate reaction feasibility more broadly. In summary, while the best way to evaluate model precision is not clear, we think this needs more attention and thought from the community.

**Best practice S2: use consistent and realistic post-processing** Most prior works perform some amount of post-processing of model outputs when measuring accuracy. Unfortunately, this has not been done consistently by previous papers, distorting comparisons between methods. In general, the evaluation post-processing should match the post-processing that would be performed if the model was used in a CASP program. We identify several instances of this below and suggest best practice.

- **Invalid outputs:** some models can output invalid molecules (e.g. a syntactically invalid SMILES string) (Irwin et al., 2022). When computing top-$k$ accuracy, some prior works include invalid molecules in the top-$k$, whereas other works filter them and consider the top-$k$ *valid* molecules. Because the validity of molecules is generally easy to check,[1] a well-engineered CASP program would discard invalid molecules instead of considering them during search. Therefore, we believe best practice should be to only consider valid molecules when computing top-$k$ accuracy.

- **Duplicate outputs:** some models can produce the same result (i.e. same set of reactants) multiple times. Clearly, a well-engineered CASP program would remove duplicate reactions, because they are redundant for search. However, this has not been done consistently in prior work. For example, we found that the published top-5 accuracy of GLN (Dai et al., 2019) on USPTO-50K can be increased by as much as 5.8% by applying simple deduplication. Therefore, we think best practice is to measure accuracy *after* deduplicating the outputs.

- **Stereochemistry:** in general, the stereochemistry of chiral molecules is important for chemical reactivity;[2] for this reason, many prior works require an exact match of stereochemistry in order for a prediction to count as correct. In popular datasets like USPTO (Schneider et al., 2016) stereochemistry is often unlabelled or mislabelled, which motivated the authors of LocalRetro (Chen & Jung, 2021) to measure a relaxed notion of accuracy where a prediction can be deemed correct even if its stereochemistry is different to the dataset.[3] However, this practice was not applied to baselines LocalRetro was compared to, and subsequent authors copied the result from (Chen & Jung, 2021) unaware that it uses a different definition of success. In our re-evaluation we found that using a relaxed comparison significantly boosted the reported accuracy of LocalRetro on USPTO-50K (e.g. +1.3% top-1 and +2.6% top-50); same is true for RetroKNN (Xie et al., 2023a) which built upon LocalRetro and re-used their evaluation code. While some datasets like USPTO-50K indeed contain chirality errors, in real-world scenarios CASP programs should not discard it; we therefore believe that best practice is to report the standard exact match (although additional reporting of results with stereochemistry removed could still provide valuable insight).

**Best practice S3: report inference time** In contemporary ML works, it is common to give little attention to inference time, and focus on pushing the quantitative model performance. However, in retrosynthesis prediction, the purpose of a single-step model is to act as an environment during multi-step search. In practice, having a drastically faster single-step model can translate to doing a much more extensive search, thus single-step model speed is directly tied to quantitative performance downstream. Due to that, we believe future research should give more attention to accurately reporting inference speed, reasoning in terms of a speed-accuracy Pareto front rather than accuracy alone. At the very least, we believe best practice is to report inference time in addition to accuracy.

**Best practice S4: focus on prediction with unknown reaction type** Most single-step works using USPTO report two sets of metrics: one for when the reaction type is not known, and another one for when the reaction type is given as auxiliary input; a practice started by (Liu et al., 2017). The rationale for the latter usually involves an interactive setting where a chemist may prefer one reaction type over another. In the context of multi-step search this information would not be available, and it is unlikely that a given reaction type is universally preferred across the entire search tree. In any case, none of the popular multi-step search algorithms add reactions conditioned on a particular reaction type, so this "conditional reaction prediction" would not be used by existing approaches. Thus, our recommendation is for researchers to focus on the "reaction type unknown" setting, as this is the one most directly applicable to multi-step search.

---

[1]In `rdkit` the `MolFromSmiles` method can evaluate the validity of a SMILES string in $< 1$ ms.

[2]*Stereochemistry* pertains to how atoms are arranged in 3D space. A molecule is *chiral* if it cannot be superimposed onto its mirror image, thus to be fully specified it needs explicit stereochemistry information.

[3]In LocalRetro a prediction is considered correct if its set of stereoisomers is either a subset or a superset of the set of stereoisomers of the ground-truth answer; see `isomer_match` in github.com/kaist-amsg/LocalRetro.

**Best practice S5: avoid data leakage through atom mappings**  Some results on USPTO-50K were later found to be flawed due to unexpected behaviour of rdkit canonicalization after removing the atom mapping from test molecules (Yan et al., 2020; Gao et al., 2022). While this problem is known to many practitioners, we mention it for completeness. To avoid this pitfall, the input molecules should be provided to the model with the atom mapping information removed, and they have to be re-canonicalized *after* said removal.

## 2.2 MULTI-STEP SEARCH

The role of a multi-step search algorithm is to use a single-step model, a set of purchasable molecules, and optionally some heuristic functions, in order to find synthesis routes. Most prior works evaluate multi-step search algorithms by reporting the fraction of test molecules solved in a given time window, where time is often measured with the number of calls to the reaction model. As in the previous section, here we explain the pitfalls and abuses of this metric and suggest best practice going forward.

**Pitfall M1: changing the single-step model**  Many algorithms use single-step reaction models not only to define the search environment, but also use the rankings or probabilities from a single-step model as a policy, cost function, or to otherwise guide the search (Segler et al., 2018; Kishimoto et al., 2019; Chen et al., 2020). Naturally this has led some works to modify the single-step model in order to improve search performance (Kim et al., 2021; Yu et al., 2022). These modifications not only change the relative rankings, but also the set of produced reactions. We see two pitfalls with the way this has been used in practice. First, unless the single-step model is separately validated, it is not clear whether it still outputs realistic reactions: for example, a change in the solution rate could just be the result of new unrealistic reactions being outputted by the model. Second, even disregarding model quality, comparing search algorithms with different single-step models is essentially comparing two algorithms in different environments, which is not a meaningful comparison. We think that best practice in this aspect should be training a policy model to *re-rank* the top-$k$ outputs of a fixed single-step model without changing the set of feasible reactions. This allows for meaningful improvement while still keeping the same accuracy guarantees and comparability of using the original single-step model. We note that this strategy was recently used by Liu et al. (2023).

**Pitfall M2: using search success rate to compare single-step models**  Some works (Hassen et al., 2022; Torren-Peraire et al., 2023) run search using various single-step models and use the success of such search to rank the models themselves. While we agree that single-step models should be benchmarked as part of search, inferring that a model is better solely because it allows for finding more routes can lead to flawed conclusions: an overly permissive single-step model may yield many routes simply because it lets search make unrealistic retrosynthetic steps, as demonstrated in the baseline experiments in Segler et al. (2018). Instead, success rate should be treated as an initial metric; a final determination of whether one end-to-end retrosynthesis pipeline is better than another is only possible if the quality of routes found is properly assessed. Outside of actually running synthesis, this could also be achieved using a reaction feasibility model; however, training such models in a generalizable way is so far an underexplored research direction.

**Best practice M3: carefully choose how search experiments are capped if varying the single-step model**  Existing works differ in how search experiments are limited: some use number of calls to the reaction model (Tripp et al., 2022), while others combine this with a wall-clock time limit (Hassen et al., 2022). Capping the number of model calls is a reliable choice if the single-step model is kept fixed; however, varying the single-step model can lead to some models being allocated vastly more *resources* (e.g. time) than others (Torren-Peraire et al., 2023). This may be justified if one believes the model speed is subject to change, and that perhaps all compared models can be optimized to eventually take a similar amount of time per call, but in the absence of such belief we recommend limiting search using a measure that treats the algorithm as a black-box (e.g. wall-clock time or memory consumption), as such approach also more directly reflects downstream use in CASP systems.

**Best practice M4: cache calls to the reaction model**  If the same molecule is encountered twice during search, a naive implementation will call the reaction model twice. As calling the reaction model is expensive, a well-engineered CASP system would clearly *cache* the outputs of the reaction model to avoid duplicate computation. Therefore, we believe it is best practice to use a cache for the single-step

model when evaluating multi-step algorithms. This may sound like a minor implementation detail, but it actually has a significant impact on the evaluation: often large sub-trees can occur in multiple places during search;[4] without a cache, expanding each occurrence of these subtrees will count against an algorithm's time budget, whereas with a cache these expansions are effectively free (Tripp et al., 2022).

**Best practice M5: evaluate the diversity of proposed routes**  While previous works emphasize finding a single synthesis route quickly, because outputs of CASP programs may not work in the wet lab it is preferable to return multiple routes, and that these routes be *diverse*. Put another way, once an algorithm is able to find a single route, it is desirable to evaluate its ability to find additional ones which differ from the one already found. There are many ways to measure diversity, but we think that a good diversity metric must be monotonic with respect to input routes (otherwise algorithms could be penalized for finding more routes). One such metric is the packing number, also called #Circles (Xie et al., 2023b), which can be instantiated as the number of synthesis routes with no overlapping reactions.[5]

**Best practice M6: quality assessment by chemists**  Finally, we recommend practitioners perform qualitative assessment of the discovered routes by expert chemists, similarly to Segler et al. (2018). This is closer to experimental validation than commonly used metrics, and has the potential to catch many pitfalls, including (but not limited to) most of those described above. Additionally, it can capture poor synthesis strategies e.g. repetitions of similar steps, redundant (de)protection chemistry, or poor choice of linear vs convergent synthesis routes, which are difficult to spot with computational metrics.

## 3 SYNTHESESUS

To encourage and promote the principles and practices discussed in Section 2, we built a benchmarking library called SYNTHESEUS. SYNTHESEUS is designed to be a platform for researchers developing methods for retrosynthesis, rather than a specific set of models or tasks. Currently, there is no generic package for retrosynthesis evaluation, forcing researchers to either write evaluation code themselves (which can be subtly inconsistent with prior work) or directly copy code from prior works (which have not followed the best practices from Section 2). SYNTHESEUS provides a working end-to-end retrosynthesis pipeline which is modular and extensible for both novel single-step models and novel multi-step search algorithms; this allows researchers to plug their methods into a well-tested evaluation pipeline which implements best practice by default. We highlight key features of SYNTHESEUS below, but refer the reader to github.com/anonymous/anonymous for an in-depth look into its API.

**Unrestricted single-step model development**  SYNTHESEUS uses a minimal standard interface to interact with single-step models. This enables users to build their models separately from SYNTHESEUS and integrate them by writing a thin wrapper, allowing SYNTHESEUS to evaluate and use all single-step models in a consistent way. Furthermore, as the framework controls the inputs and outputs of the wrapped model, it automatically prevents "cheating" in the form of relying on atom mappings (S5), takes care of post-processing the outputs when evaluating accuracy (S2), and measures inference time (S3). When used in multi-step search, SYNTHESEUS also automatically performs caching (M4).

**Separation of components in multi-step search**  SYNTHESEUS cleanly separates the various components of a multi-step search algorithm: the single-step model, set of purchasable molecules, the search graph, and search heuristics (policies and value functions). This makes it easy to change one part of a CASP program and see the effect: for example, run MCTS with two different single-step models, or run Retro* with two different sets of purchasable molecules.

**Detailed metrics for multi-step search**  In addition to tracking whether a synthesis route has been found, SYNTHESEUS also tracks *when* it has been found using several different time measures (wallclock time, number of calls to the reaction model), making it easy to track the performance of an algorithm over time. SYNTHESEUS also implements several diversity metrics (M5), and provides visualization tools to allow the routes to be inspected by researchers or expert chemists (M6).

---

[4]For example, if the reactions $M \rightarrow A + B$ and $M \rightarrow A + C$ are possible, then any subsequent reactions on molecule $A$ will be repeated multiple times.

[5]Finding the largest set of non-overlapping routes is equivalent to the set packing problem, which is NP-hard (Karp, 2010). In practice there is no need to compute it exactly, and a heuristic approximation is sufficient.

## 4 EXPERIMENTS: RE-EVALUATION OF EXISTING METHODS

We use SYNTHESEUS to re-evaluate many existing single-step models in conjunction with popular search algorithms, providing a holistic view of the existing methods, and in many instances *correcting the numbers from the literature*. We did not re-implement any of the models, and used open-source codebases wrapped into SYNTHESEUS's single-step model interface, demonstrating its flexibility. Crucially, *results in this paper were produced by our evaluation framework with no numbers copied from previous work*, which ensures a fair comparison immune to many issues discussed in Section 2.

### 4.1 SINGLE-STEP

**Datasets**   As a starting point we use the USPTO-50K dataset (Schneider et al., 2016) split by Dai et al. (2019), as all of the models we consider report results on this dataset, allowing us to contrast the published numbers with ones obtained from our re-evaluation. There is a newer version of this dataset available (Lin et al., 2022), but since our aim is to correct the existing results, we focus on the more established version and leave using the newer one for future work. Moreover, USPTO-50K is a small dataset, and it may not be representative of the full data distribution. Thus, we also use the proprietary Pistachio dataset (Mayfield et al., 2017) (more than 15.6M raw reactions; 3.7M samples after preprocessing), and evaluate out-of-distribution generalization of the model checkpoints trained on USPTO-50K. To the best of our knowledge this has not been done before; while some works also make use of Pistachio (Jiang et al., 2023), it is rather used as a *pretraining* dataset before fine-tuning on USPTO. As most researchers do not have access to Pistachio, by reporting generalization we aim to gain insight into how USPTO-trained models work across a wider input distribution they may be exposed to during multi-step search. We performed significant cleaning and preprocessing on the Pistachio data to ensure the test set is of high quality, and also to limit overlap between the USPTO training set and Pistachio test set; see Appendix A for the details of our preprocessing pipeline.

**Models**   We re-evaluate established single-step models where either the code is publicly available (GLN (Dai et al., 2019), MEGAN (Sacha et al., 2021), MHNreact (Seidl et al., 2021), LocalRetro (Chen & Jung, 2021), RootAligned (Zhong et al., 2022) and Chemformer (Irwin et al., 2022)) or we were able to obtain it from the authors (RetroKNN (Xie et al., 2023a)). We omit Dual-{TB,TF} (Sun et al., 2020) as we have no access to the code; even though these models reported promising performance, we were unable to verify it under our framework. For all models we used the provided checkpoint if one using the right data split was available, and trained a new model using the original training code otherwise. We used the original implementations adapted to our shared interface.[6]

**Metrics**   We compute top-$k$ accuracy for $k$ up to $50$ and Mean Reciprocal Rank (MRR). It is not clear what value of $k$ is the most relevant metric to consider, but given the target use of single-step models in search, it is desirable for $k$ to be roughly similar to the expected or desired breadth of the search tree (number of children visited for a typical internal node); thus, $k = 1$ would be too narrow. Typically, values beyond $k = 50$ are not reported, as models tend to saturate past this point. Several CASP programs also restrict the expansion beyond the top-50 (Segler et al., 2018; Genheden et al., 2020). We highlight $k = 5$ as a reasonable middle-ground, and defer extended results to Appendix B.

**Setup**   We queried all models for $n = 100$ outputs (see Appendix C for a discussion on how obtaining multiple results is handled for different model types). Note that we measure top-$k$ only up to $k = 50$ but set $n > k$ to account for deduplication. We used a fixed batch size of 1 for all models. While all models could easily handle larger batches, batch size used during search typically cannot be set arbitrarily, and in most cases it is equal to 1 as usually search is not parallelized. Thus, speed under batch size of 1 directly translates to the maximum number of model calls that can be performed during search with a fixed time budget. All inference time measurements used a single V100 GPU.

**Results**   We present top-5 accuracy results on both datasets in Figure 1. First, we note that two of the models (RootAligned, Chemformer) predict the reactants SMILES from scratch using a Transformer decoder (Vaswani et al., 2017), while the other models predict the graph rewrite to apply to the product.

---

[6]Notably, when adapting MHNreact we found that its use of multiprocessing was highly suboptimal; our wrapped version partially fixes this issue and is thus more performant than the original code.

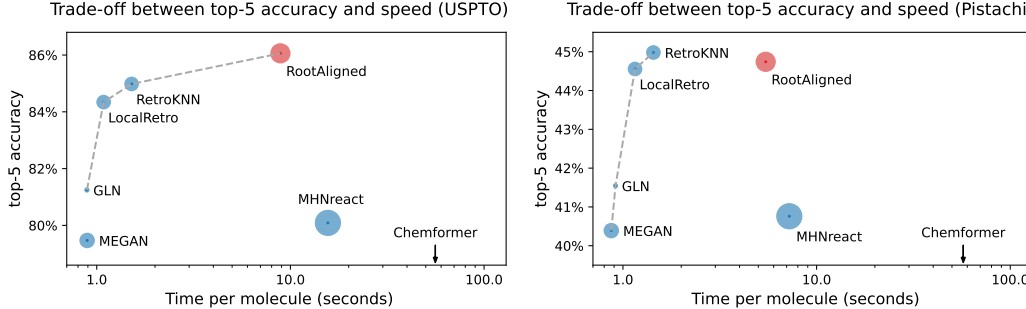

Figure 1: Trade-off between top-5 accuracy and inference speed. Circle area is proportional to the number of parameters; color denotes whether a model predicts a graph edit (blue) or produces the output from scratch (red). Dashed gray line shows the Pareto front (best result for any time budget). Exact results for Chemformer are not shown as they fall below the plot boundary. We show in-distribution results on USPTO-50K (left) and out-of-distribution generalization on Pistachio (right).

Across datasets and metrics, models of the former type tend to be slower, and while they show good performance in top-1 accuracy, they are surpassed by the graph-transformation-based models for higher $k$. We hypothesize that, due to more explicit grounding in the set of transformations occurring in training data, transformation-based models tend to produce a more complete coverage of the data distribution. Second, many of the USPTO-50K results we report are better than the numbers from the literature (see Table 1 in Appendix B for a detailed breakdown), especially in terms of top-$k$ accuracy for $k > 1$, which is affected by deduplication. This also changes some of the model rankings, e.g. LocalRetro was originally reported to have a better top-1 accuracy than GLN, but we find that to not be the case. Surprisingly, model ranking on USPTO-50K transfers to Pistachio quite well, although all results are substantially degraded, e.g. in terms of top-50 accuracy all models still fall below $55\%$, compared to nearly $100\%$ on USPTO. While for template-based models this is a result of insufficient coverage, we note that some of the models tested here are template-free, and yet they fail to generalize better than their template-based counterparts (this is similar to the findings of Tu et al. (2022)). To further ground our Pistachio results, we note that Jiang et al. (2023) report $66.1\%$ top-5 accuracy when training on Pistachio directly (compared to our transfer results of up to $45\%$); however, these values are not fully comparable due to differences in preprocessing. Finally, RetroKNN is best or close to best on all metrics on both datasets, while also being one of the faster models in our re-evaluation. However, we caution the reader to not treat this as a final recommendation; as discussed in Section 2, existing single-step metrics provide a useful but incomplete view of the performance of single-step models.

## 4.2 MULTI-STEP

We also ran search experiments combining various single-step models and search algorithms. As our primary objectives are to outline good practices, correct established numbers, and showcase SYNTHE-SEUS, we only show preliminary multi-step results; we leave a final determination of which end-to-end pipeline is best to future work building on top of our framework. It is worth noting that the single-step models considered here use various (usually conflicting) versions of deep learning frameworks and other libraries, yet due to minimalistic dependencies SYNTHESEUS can be combined with any of them.

**Setup** We followed one of the experimental setups from Tripp et al. (2022) and used the 190 target molecules from the Retro* Hard set (Chen et al., 2020). We combined each of the seven single-step models with two search algorithms: an MCTS variant and Retro* (Chen et al., 2020). All runs were limited to 10 minutes per molecule. Our single-step model wrappers expose the underlying output probabilities, which are used by both algorithms to guide the search. To ensure a fair comparison, the hyperparameters of each algorithm-model combination were tuned separately (see Appendix D for the exact procedure; qualitatively this step was especially important for MCTS).

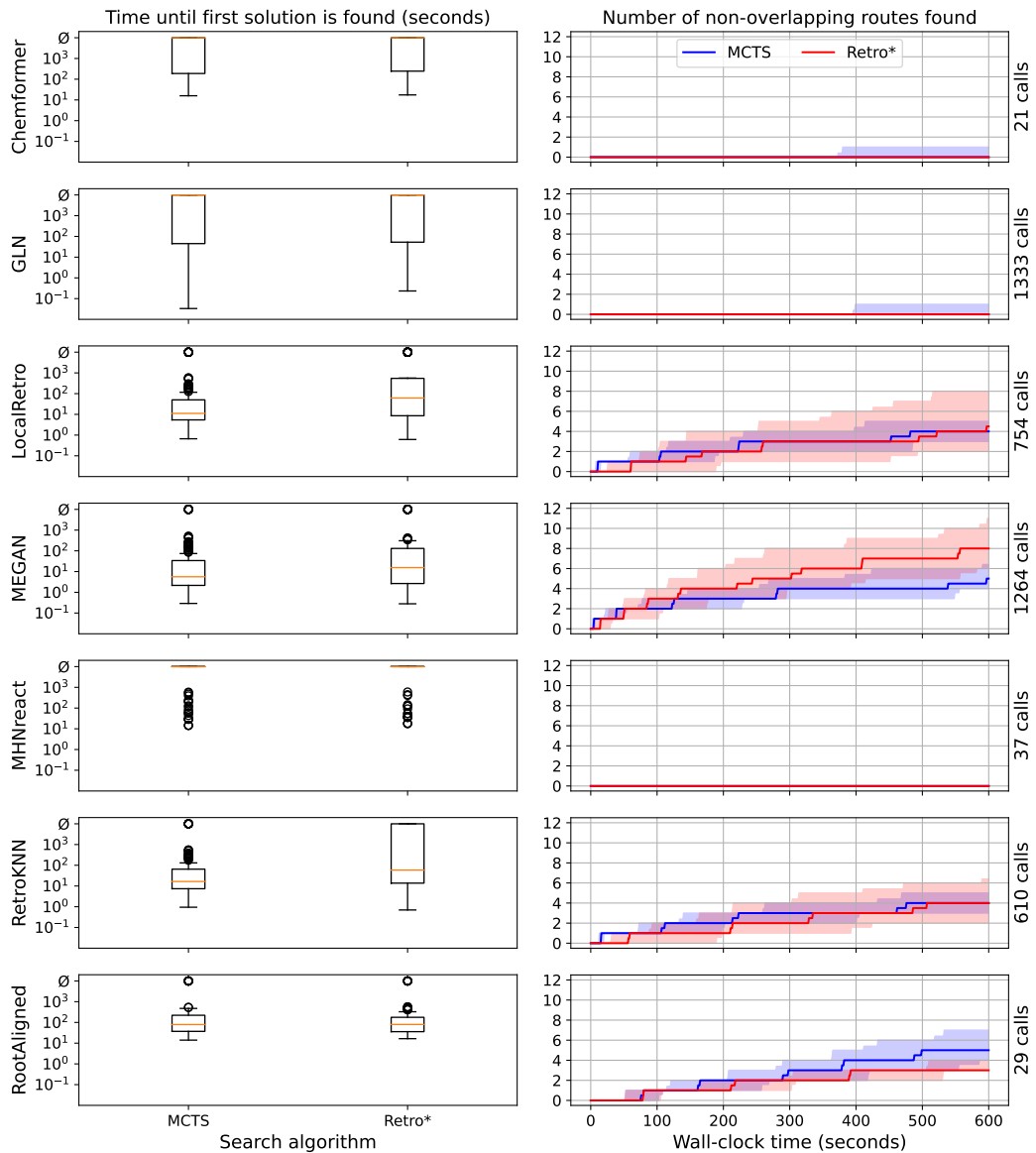

Figure 2: Multi-step search results on the Retro* Hard target set with different single-step models. **Left:** Time until first solution was found (or ∅ if a molecule was not solved). Orange line represents the median, box represents 25th and 75th percentile, whiskers represent 5th and 95th percentile, points outside this range are shown as dots. **Right:** Approximate number of non-overlapping routes present in the search graph (tracked over time and aggregated across target molecules). Solid line represents the median, shaded area shows the 40th and 60th percentile. On the right hand side we note the average number of calls made by the model within the allotted time limit.

**Results** We show the results in Figure 2, tracking when the first solution was found as well as the maximum number of non-overlapping routes that can be extracted from the search graph. For LocalRetro, MEGAN, RetroKNN and RootAligned, both search algorithms are able to find several disjoint routes for most molecules, while for other models search can only solve a minority of the targets. Notably, RootAligned obtains promising results despite making less than 30 calls on average (due to its high computational cost). However, as discussed in Section 2.2, these results should not be treated as a final comparison of the models and rather serve as a starting point towards future research.

## 5    RELATED WORK

Many works have proposed benchmarks for retrosynthesis. For single-step, the USPTO-50K dataset (Schneider et al., 2016) is the most popular, while for multi-step many papers report results on the 190 hard molecules from Retro* (Chen et al., 2020). However, these benchmarks do not have a standardized evaluation pipeline, leading to inconsistent re-implementations by different authors often subject to pitfalls discussed in Section 2 (particularly S2, S5 and M1, M4). SYNTHESEUS allows these benchmarks to be run in a consistent and comparable way. Nonetheless, these benchmarks are far from perfect: their standard metrics include recall (S1) and success rate (M2) but do not include inference time (S3) or diversity (M5). In a major step forward, Genheden & Bjerrum (2022) propose the PaRoutes benchmark for multi-step search, which does include an assessment of diversity (M5) and a standardized evaluation script. Unfortunately, it measures diversity with the output of a clustering algorithm: a metric which is non-monotonic. This makes it possible for an algorithm to find strictly more routes than another algorithm yet be rated as less diverse. In contrast, the diversity metrics included with SYNTHESEUS are monotonic, meaning that finding additional routes will never cause diversity to decrease.

More broadly, some works have highlighted the deficiencies with retrosynthesis evaluation. Zhong et al. (2023) point out how separate evaluation of single-step and multi-step may not lead to effective CASP programs and mention the limitations of recall (S1). Segler et al. (2018) noted the inherent shortcomings of *in-silico* evaluation and benchmarked their algorithm with an A/B test by expert chemists. However, evaluation through human feedback is not scalable, making such examples rare in ML venues. Hassen et al. (2022) correctly noted that the performance of multi-step search algorithms will depend on the single-step model and performed a large evaluation of many single-step models combined with popular multi-step search algorithms; this analysis was later extended in Torren-Peraire et al. (2023) to large proprietary training datasets. However, these works quantitatively compare the results across different single-step models using success rate, which we argue is not best practice (M2).

Finally, it is worth mentioning several popular software packages for retrosynthesis. ASKCOS (Coley et al., 2019) and AiZynthFinder (Genheden et al., 2020) are software packages for multi-step search with a simple interface and interactive visualizations. However, they are primarily designed to support MCTS with template-based models. In contrast, SYNTHESEUS is designed in a model-agnostic and algorithm-agnostic way, and is easy to extend to arbitrary models and algorithms. IBM RXN (rxn.res.ibm.com) and Chematica (Klucznik et al., 2018) are popular software tools for retrosynthesis, but unlike SYNTHESEUS cannot be used for benchmarking as they are closed-source. Ultimately, none of these packages act as benchmarking platforms to the same degree as SYNTHESEUS.

## 6    CONCLUSION AND FUTURE WORK

In this paper we presented an analysis of pitfalls and best practices for evaluating retrosynthesis programs (Section 2), a software package called SYNTHESEUS to help researchers benchmark their methods following these best practices (Section 3), and used SYNTHESEUS to re-evaluate many existing models and algorithms (Section 4). These results "set the record straight" regarding the performance of existing algorithms, and the standardized evaluation protocol of SYNTHESEUS can ensure that future works do not continue to make the same mistakes. We encourage members of the community to contribute new models, algorithms, and metrics to SYNTHESEUS (see maintenance plan in Appendix E).

Despite this, several important issues remain in the field, which we plan to resolve with SYNTHESEUS in future iterations. As we argue in Section 2, existing metrics of recall (S1) and solve rate (M1/M2) are not ideal for comparing arbitrary end-to-end retrosynthesis pipelines. Assuming evaluation by chemists (M6) is not possible, we believe the most plausible substitute is to develop reaction "feasibility" models to estimate whether reactions will succeed. If such models were used post-hoc (not available during training or search), they could be used to evaluate the precision of single-step models (resolving S1) and assign a feasibility score to entire routes (resolving M1/M2). We designed SYNTHESEUS with this in mind and have a clear way to support feasibility models in both single-step and multi-step evaluation. However, how to train a high-quality feasibility model is an open research question which we leave to future work. Finally, the lack of information on reaction conditions, required quality of the starting materials, required equipment, and purification, is a significant barrier to actually executing the synthesis plans from CASP systems which SYNTHESEUS does not address. We encourage the community to work together with us on these challenges in the future.

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

## A PISTACHIO PREPROCESSING

The raw Pistachio data (version 2023Q2, released in June 2023) contained $15\,684\,711$ raw reactions; however, this included many duplicates, outliers (e.g. reactions with extremely large products), and potentially samples that are erroneous. To ensure the test data is of high quality, we performed careful filtering and processing of the raw Pistachio data. We applied the following steps in order:

- Remove duplicate reactions.
- Remove reactions with more than $4$ reactants.
- Compute occurrence count of each product molecule across the dataset (counting individual products in multi-product reactions separately). For every reaction with products $[p_1, ..., p_m]$ (including reactions with a single product i.e. $m = 1$), remove all side products. Product $p_i$ is considered a side product if it either has less than $5$ atoms, or appears at least $1000$ times across the dataset. The latter condition allows us to remove common side products, which may have $5$ or more atoms but are still uninteresting. Retain only those reactions where exactly one $p_i$ remained after this procedure (i.e. those with a well-defined main product).
- Remove reactions where the (now unique) product has more than $100$ atoms.
- Remove reactions where the ratio of the number of reactant atoms to the number of product atoms exceeds $20$.
- Remove reactions where the product appears as one of the reactants.
- Remove reactions where the product contains more than 3 atom types which do not appear in the reactants, or more than 3 atoms whose atom mapping IDs do not appear in reactants.[7]
- Remove reactions with missing or erroneous atom mapping.
- Refine reactions by removing reactants that do not contribute atoms to the product.

We chose the processing steps above such that we exclude erroneous reactions, extreme outliers (i.e. those that are either very large or have an extreme imbalance between the size of the reactants and the size of the product), and reactions with no clearly defined main product. These processing steps (and the particular constants used therein) were informed by expert qualitative analysis of the reactions, as well as practical considerations. For example, we found that several very large outliers in raw Pistachio data seem to cause `rdkit`'s template extraction routines to hang; however, these reactions did not survive our filtering.

After the preprocessing we obtained $3\,690\,362$ single-product samples, which we grouped by their product, and split into train, validation and test sets following a 90/5/5 ratio, making sure the groups of samples with the same product are placed into the same fold. We used a random split, except for those products which were found in USPTO-50K data; in those cases, we attempt to place the corresponding group of samples in the same fold as it appears in USPTO (this limits overlap between training set of one dataset and test set of another, which could distort our generalization results). As the USPTO-50K split from Dai et al. (2019) contains a small amount of product overlap between folds, this process of ensuring a "compatible" Pistachio split was imperfect; the product overlap between USPTO-50K training set and Pistachio test set is non-zero, but it is negligibly small. Note that an alternative approach to preventing overlap would be to completely remove USPTO products from Pistachio before splitting the dataset, but we did not want to artificially exclude (valuable) products present in USPTO-50K.

In this work we use Pistachio solely for testing generalization, thus we only used the test fold, which we randomly subsampled to $20\,000$ samples for faster evaluation (note that this is still 4 times larger than the test set of USPTO-50K). We described the full procedure to generate all folds to facilitate future work.

---

[7]This step may appear very specialized; indeed, it was chosen to target some of the remaining erroneous samples, with only 121 reactions removed by this filter.

# B    EXTENDED SINGLE-STEP RESULTS

## B.1    TRADE-OFF BETWEEN QUANTITATIVE PERFORMANCE AND SPEED

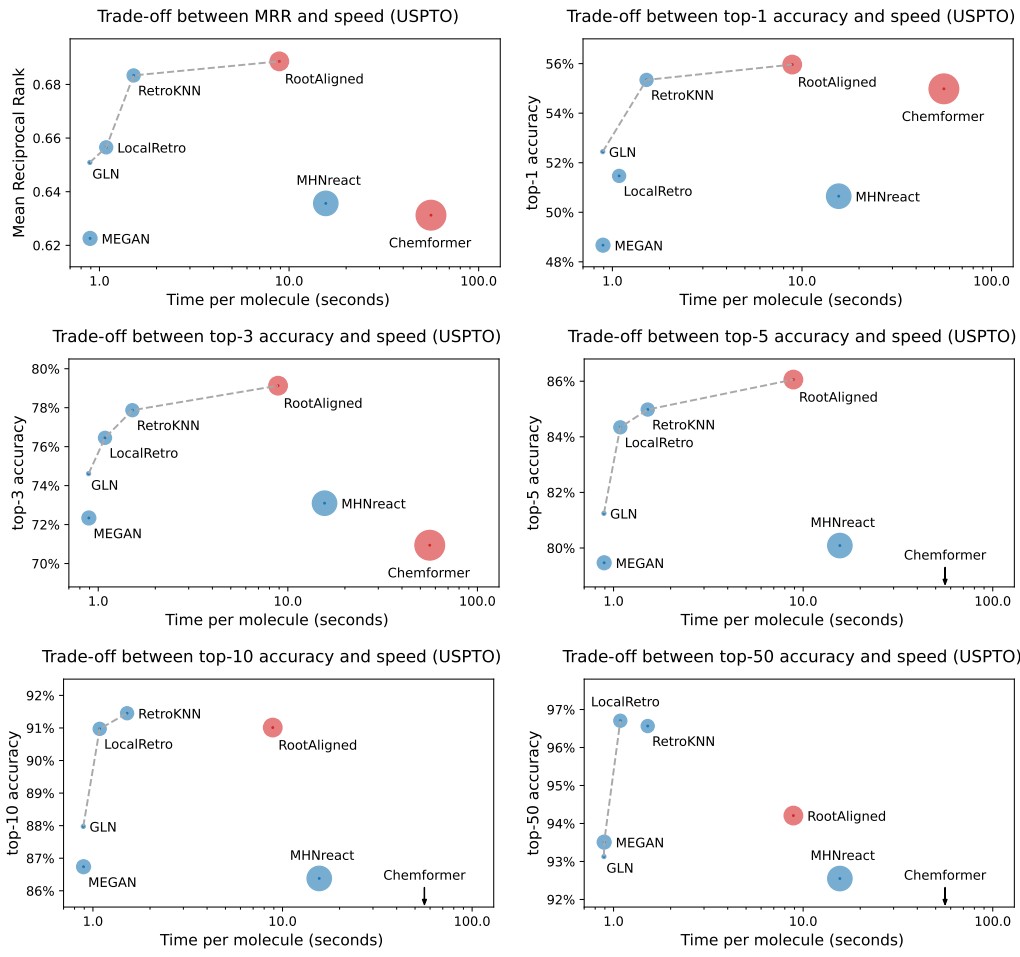

Figure 3: Results on USPTO-50K in same format as Figure 1 but extended with top-1, top-3, top-10, top-50, and MRR. Plot for top-5 shown in Figure 1 is reprinted here for convenience.

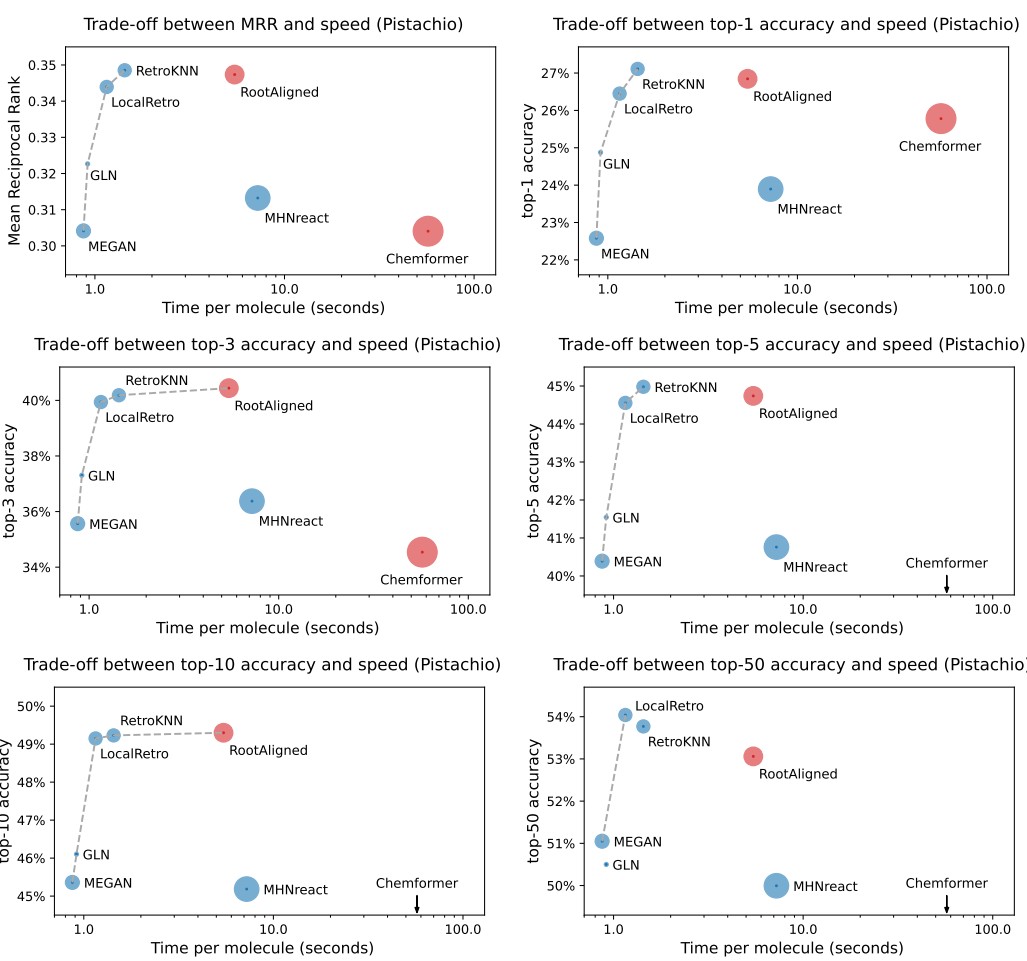

Figure 4: Results on Pistachio in same format as Figure 1 but extended with top-1, top-3, top-10, top-50, and MRR. Plot for top-5 shown in Figure 1 is reprinted here for convenience.

## B.2 Comparison with Published Results

In Table 1, we present the results from Figure 3 in numeric form, as well as contrast them with the published numbers. For results produced with SYNTHESEUS we additionally investigate the effect of deduplication.

Table 1: Results on USPTO-50K compared to the numbers reported in the literature. SYNTH denotes whether we used SYNTHESEUS to produce the result (as opposed to copying the published number, or, in case of LocalRetro with exact match, generating the number ourselves using authors' code), D denotes whether deduplication was performed (which in SYNTHESEUS is enabled by default, but can be turned off). We underline values that differ significantly from the previous row (at least $0.7\%$ for top-$k$ or $0.003$ for MRR), and use colors to distinguish whether the value is better (green) or worse (red) than the row directly above.

| Model | SYNTH | D | top-1 | top-3 | top-5 | top-10 | top-50 | MRR |
|---|---|---|---|---|---|---|---|---|
| GLN | ✗ | | 52.5% | 69.0% | 75.6% | 83.7% | 92.4% | - |
| | ✓ | ✗ | 52.4% | 68.8% | 75.4% | 83.5% | 92.5% | 0.6262 |
| | ✓ | ✓ | 52.4% | 74.6% | 81.2% | 88.0% | 93.1% | 0.6509 |
| MEGAN | ✗ | | 48.1% | 70.7% | 78.4% | 86.1% | 93.2% | - |
| | ✓ | ✗ | 48.7% | 71.9% | 78.9% | 86.0% | 93.2% | 0.6203 |
| | ✓ | ✓ | 48.7% | 72.3% | 79.5% | 86.7% | 93.5% | 0.6226 |
| MHNreact | ✗ | | 50.5% | 73.9% | 81.0% | 87.9% | 94.1% | - |
| | ✓ | ✗ | 50.6% | 73.1% | 80.1% | 86.4% | 92.6% | 0.6356 |
| | ✓ | ✓ | 50.6% | 73.1% | 80.1% | 86.4% | 92.6% | 0.6356 |
| LocalRetro (exact match) | ✗ | | 53.4% | 77.5% | 85.9% | 92.4% | 97.7% | - |
| | ✗ | ✗ | 52.0% | 75.5% | 83.4% | 90.0% | 95.7% | - |
| | ✓ | ✗ | 51.5% | 75.6% | 83.5% | 90.6% | 96.7% | 0.6530 |
| | ✓ | ✓ | 51.5% | 76.5% | 84.3% | 91.0% | 96.7% | 0.6565 |
| RootAligned | ✗ | | 56.3% | 79.2% | 86.2% | 91.0% | 94.6% | - |
| | ✓ | ✗ | 56.0% | 79.1% | 86.1% | 91.0% | 94.2% | 0.6886 |
| | ✓ | ✓ | 56.0% | 79.1% | 86.1% | 91.0% | 94.2% | 0.6886 |
| Chemformer | ✗ | | 54.3% | - | 62.3% | 63.0% | - | - |
| | ✓ | ✗ | 55.0% | 67.8% | 70.5% | 72.5% | 74.8% | 0.6182 |
| | ✓ | ✓ | 55.0% | 70.9% | 73.7% | 75.4% | 76.0% | 0.6312 |
| RetroKNN | ✗ | | 57.2% | 78.9% | 86.4% | 92.7% | 98.1% | - |
| | ✓ | ✗ | 55.3% | 76.9% | 84.3% | 90.8% | 96.5% | 0.6796 |
| | ✓ | ✓ | 55.3% | 77.9% | 85.0% | 91.5% | 96.6% | 0.6834 |

Focusing on the most significant differences between the results, we make the following observations:

- GLN's published results match those obtained with SYNTHESEUS with no deduplication. However, its top-$k$ accuracies for $k > 1$ improve significantly with deduplication turned on.

- MEGAN's published results improve slightly after moving to SYNTHESUES, and then there is a small further improvement from deduplication. We hypothesize the former might be a result of retraining the model (while the authors did release a checkpoint trained on USPTO-50K, our analysis seemed to indicate that model used a different data split for training, as the performance on our USPTO-50K test set was unrealistically high).

- MHNreact's results are not affected by deduplication, but the numbers we obtain with SYNTHESEUS are worse than those originally published; this may be explained by either the fact that we retrained the model or implementation details.

- LocalRetro (and by extension RetroKNN) used a relaxed notion of success, and we see that the results deteriorate significantly when using SYNTHESEUS. For LocalRetro, we additionally measured accuracy using authors' original code but replacing the relaxed match with an exact one (see row labelled with "(exact match)"), which caused a similar drop in

performance, confirming that the way of measuring accuracy is indeed responsible for the difference. Both LocalRetro and RetroKNN improve due to deduplication, but the final results still fall short of the originally reported numbers.

- RootAligned's published results closely match those obtained with SYNTHESEUS and are unaffected by deduplication, showing this model likely already conforms to many of the best practices from Section 2.

- Finally, Chemformer's results are improved when switching to SYNTHESEUS, and then further when turning on deduplication. The former could be explained by the fact that SYNTHESEUS removes invalid molecules, which Chemformer (as a SMILES-based model) can produce.

Next, in Table 2 we present the exact numbers corresponding to the results from Figure 4. However, here we cannot compare to published results, as to the best of our knowledge these are not available.

Table 2: Generalization results on Pistachio in numeric form.

| Model | top-1 | top-3 | top-5 | top-10 | top-50 | MRR |
|---|---|---|---|---|---|---|
| GLN | 24.9% | 37.3% | 41.5% | 46.1% | 50.5% | 0.3227 |
| MEGAN | 22.6% | 35.6% | 40.4% | 45.4% | 51.0% | 0.3041 |
| MHNreact | 23.9% | 36.4% | 40.8% | 45.2% | 50.0% | 0.3132 |
| LocalRetro | 26.5% | 39.9% | 44.6% | 49.1% | 54.0% | 0.3439 |
| RootAligned | 26.8% | 40.4% | 44.7% | 49.3% | 53.1% | 0.3473 |
| Chemformer | 25.8% | 34.5% | 36.5% | 37.9% | 38.5% | 0.3041 |
| RetroKNN | 27.1% | 40.2% | 45.0% | 49.2% | 53.8% | 0.3485 |

## C  OBTAINING MULTIPLE RESULTS FROM SINGLE-STEP MODELS

During evaluation, we need to obtain $n$ results for a given input. It is worth noting that the value of $n$ is used differently depending on model type: models based on templates and local templates (GLN, MHNreact, LocalRetro and RetroKNN) first process the input and then apply the templates until $n$ results are obtained, while models that employ a sequential auto-regressive decoder (MEGAN, Chemformer) use beam search with $n$ beams. These two approaches lead to different scaling, as in the former case the bulk of the computation is amortized and does not scale with $n$, while in the latter case the entire procedure scales with $n$ essentially linearly. Finally, the RootAligned model is a special case, as it uses a combination of beam search and test-time data augmentation; scaling up either of these hyperparameters increases inference time and number of results, but the right balance between them requires careful tuning. In our work we used the default settings (20 augmentations, 10 beams) which correspond to a maximum of $20 \cdot 10 > n$ results being generated (recall that $n = 100$).

## D  SEARCH ALGORITHMS HYPERPARAMETER TUNING

To ensure a fair comparison, we tuned the hyperparameters of both MCTS and Retro* separately for each single-step model. For both algorithms we focused on tuning the component that directly interacts with the single-step model: policy in MCTS and cost function in Retro*. Notably, we did not vary many of the other components of the algorithms (e.g. reward function in MCTS or value function in Retro*) to avoid an infeasibly large search space.

All tuning runs used 25 targets from the ChemBL Hard set used in Tripp et al. (2022) and searched under a time limit of 5 minutes. As the primary objective we used the solve rate (i.e. number of solved targets), breaking ties first by the median and then mean number of non-overlapping routes found (formally, these three objectives were combined with weights 1.0, 0.1 and 0.01, respectively). For each search algorithm and single-step model combination we ran 50 trials using the default tuning algorithm in `optuna` (Akiba et al., 2019) to maximize the combined score.

For MCTS, we tuned the clipping range for the single-step model probabilities (lower bound in $[10^{-11}, 10^{-10}, ..., 10^{-5}]$, upper bound in $[0.9999, 0.999, 0.99, 0.9]$), temperature applied to the

clipped distribution (in $[0.125, 0.25, ..., 4.0, 8.0]$), bound constant (in $[1, 10, 100, 1000, 10000]$) and node value constant (in $[0.25, 0.5, 0.75]$). For Retro*, we only tuned the clipping range (over the same values as for MCTS), as the temperature would have no effect due to using a constant-0 value function (referred to as Retro*-0 in Chen et al. (2020)).

In general, we found that the single-step probability clipping range has little effect on the algorithms, and so the performance of Retro* was not significantly improved through the hyperparameter tuning. Conversely, in MCTS parameters such as bound constant and temperature can have a sizable effect on the behaviour, and indeed choosing them carefully improved performance with respect to an initial guess. While MCTS seemingly performed worse than Retro* when using untuned hyperparameters, carefully setting the parameters of the former led it to perform on par with Retro*, echoing the conclusions from Tripp et al. (2022).

# E   MAINTENANCE PLAN FOR SYNTHESEUS

We intend to actively continue and support the development of SYNTHESEUS going forward, including adding new features, fixing any bugs, and improving documentation. As SYNTHESEUS is an open-source project on GitHub, anybody is free to modify and propose changes by raising an issue or opening a pull request. We are committed to promptly responding to and engaging with all issues and pull requests.

The code to reproduce all experimental results (apart from those utilizing the proprietary Pistachio dataset) is publicly available.

