# OpenReview forum: "Re-evaluating Retrosynthesis Algorithms with Syntheseus"
_ICLR.cc/2024/Conference — Submitted to ICLR 2024_

### Official Review · Reviewer_JYEf · 2023-11-02

**Soundness:** 4 excellent
**Presentation:** 3 good
**Contribution:** 3 good
**Rating:** 8
**Confidence:** 4

**Summary:**

This paper analyzed pitfalls and best practices for evaluating retrosynthesis programs and a software package SYNTHESEUS to help researchers benchmark their methods following these best practices. SYNTHESEUS provides a standardized evaluation pipeline. The authors further re-evaluated many existing models and algorithms using SYNTHESEUS to correctly and fairly compare these methods and found the ranking of state-of-the-art models changes when evaluated carefully.

**Strengths:**

1. This paper is well-motivated and presents strong significance. It provides a working end-to-end retrosynthesis pipeline that implements best practices by default. It is true that many existing methods handle evaluation pipelines themselves, and inconsistencies in the evaluation pipelines make the comparison of different methods unreliable. For example, as the authors pointed out, LocalRetro measured a relaxed notion of accuracy where a prediction can be deemed correct even if its stereochemistry is different from the dataset, while the baseline methods do not. This makes the comparison of LocalRetro and baseline methods unfair and unreliable. The contribution is significant since the community indeed needs a well-established evaluation pipeline for a fair evaluation.

2. The authors examined the previous retrosynthesis works carefully and pointed out the shortcomings of different methods in this practice. They also suggest the best evaluation practices via the Python package SYNTHESEUS which supports consistent evaluation of single-step and multi-step retrosynthesis algorithms, with best practices enforced by default.

3.  The authors further re-evaluate existing methods, both single-step and multi-step retrosynthesis, with the proposed SYNTHESEUS pipeline and show the real effectiveness of different methods.

**Weaknesses:**

1. Some important baseline methods are not included in Figure 1, such as [1, 2] which show strong performance and also provide source code at https://github.com/uta-smile/RetroComposer and https://github.com/Jamson-Zhong/Graph2Edits, respectively.

2. Suggest careful considerations on the suggested post-processing procedures. Is it true the best practice should be to only consider valid molecules when computing top-k accuracy? I agree with the authors that invalid molecules can be removed easily. However, the ability to generate valid molecules is also important. Otherwise, the model might need to generate multiple invalid molecules before a valid molecule, which consumes more time compared to the case where the first generated molecule is valid. Same as the duplicated outputs.


[1] Yan, Chaochao, et al. "RetroComposer: Composing Templates for Template-Based Retrosynthesis Prediction." Biomolecules 12.9 (2022): 1325.

[2] Zhong, Weihe, Ziduo Yang, and Calvin Yu-Chian Chen. "Retrosynthesis prediction using an end-to-end graph generative architecture for molecular graph editing." Nature Communications 14.1 (2023): 3009.

**Questions:**

The motivation and significance of this paper are good. The only question would be the quality and ease of the Python package SYNTHESEUS. While some suggested practices are indeed important and help to remove unfair evaluation, I would suggest keeping some practices optional and configurable, such as removing invalid or duplicate outputs.

---

> ### Author Response · Authors · 2023-11-15
> **Response**
>
> **(Q1)**: RetroComposer [1] and Graph2Edits [2] should be included.
>
> **(A1)**: Thank you for the suggestion; we recently learned about Graph2Edits and were considering integrating it into `syntheseus`. We have now confirmed that the model works under our default single-step environment, and wrote a wrapper aligning it with the shared model interface. We will post precise evaluation results soon and then add them to Figure 1 of our paper.
>
> ---
>
> **(Q2)**: There should be an option to turn off removal of invalid molecules / deduplication.
>
> **(A2)**: Deduplication can be turned off by setting `skip_repeats=False` in the evaluation script. We don’t have an option to turn off the removal of invalid molecules, as the results are returned as `Molecule` objects which are guaranteed to be valid (this makes a lot of the code more robust/safe as it doesn’t have to check for invalid SMILES). However, it is possible to assess the fraction of valid predictions by comparing how many results were returned vs how many were requested. So both the fraction of valid results as well as the amount of duplicates can be computed using `syntheseus`.
>
> ---
>
> **(Q3)**: Is it truly best practice to remove invalid/duplicate molecules? The ability to generate valid/unique molecules is important as otherwise the model may take a long time before a valid/unique result can be obtained.
>
> **(A3)**: Indeed, measuring the fraction of valid/unique results is useful, and (as we mention above), this can be computed using the model wrappers present in `syntheseus`. For end-to-end evaluation, we recommend discarding invalid molecules and duplicates; if a model is asked for e.g. 50 results, but due to this filtering it is only left with 10, then this will be reflected in it’s top-k accuracy for `k > 10`, we do not re-query the model to provide more outputs until 50 are gathered. However, if desired, it is simple to implement this re-querying procedure using our `ReactionModel` abstraction. `syntheseus` does record the total time taken by model calls (something previous works usually did not account for), making it is easy to quantify the effect of such adaptive inference on model speed.

---

> > ### Author Response · Authors · 2023-11-16
> > **Graph2Edits evaluation**
> >
> > We finished testing Graph2Edits through our `syntheseus` wrapper on USPTO-50K test set. The results in comparison to the originally reported numbers are as follows:
> >
> > || top-1 | top-3 | top-5 | top-10 | top-50 |
> > |---|---|---|---|---|---|
> > | original paper | 55.1% | 77.3% | 83.4% | 89.4% | 92.7% |
> > | with `syntheseus` | 54.6% | 76.6% | 82.8% | 88.7% | 91.7% |
> >
> > We found that the results in the re-evaluation are roughly similar but slightly worse (by around 0.6%) than those reported in the original paper. We will include these results in our manuscript but first take a closer look to see if there is a clear explanation for the small differences in performance.

---

> > ### Comment · Reviewer_JYEf · 2023-11-17
> > **optional removal of invalid molecules**
> >
> > Thanks for the clarification!
> >
> > The optional removal of invalid molecules is important because it can better reflect the ability of the model, more than the fraction of valid/unique results. For example, a model is asked for the top 10 results, one model generates 9 invalid molecules until the last valid and desired molecule and its top-1 accuracy is 100%, while another model generates the first molecule which is valid and desired so its top-1 accuracy is also 100%. However, in this case, current metrics can not reflect this difference since both models have the same top-1 accuracy.

---

> ### Comment · Reviewer_JYEf · 2023-11-17
> **RetroComposer results**
>
> Thanks to the authors for adding the results of Graph2Edits! I also look forward to RetroComposer results, which is a totally different type of model that re-scores the prediction and it also achieves very competitive results.

---

> > ### Author Response · Authors · 2023-11-18
> > **RetroComposer**
> >
> > Indeed, we are also looking at RetroComposer. Integrating Graph2Edits was quicker, as the open-source repository already provides a trained checkpoint.

---

### Official Review · Reviewer_cznK · 2023-11-03

**Soundness:** 2 fair
**Presentation:** 3 good
**Contribution:** 2 fair
**Rating:** 3
**Confidence:** 4

**Summary:**

This paper introduces SYNTHE-SEUS, a benchmarking library created to evaluate the performance of both single-step and multi-step retrosynthesis algorithms. A number of previous retrosynthesis algorithms were re-evaluated on SYNTHE-SEUS. The motivation was to address the lack of standards for evaluating and comparing AI-based retrosynthesis algorithms. The experimental results reveal that when these previous algorithms are assessed on SYNTHE-SEUS under uniform conditions—identical pipelines, post-processing settings, and metrics—there are notable performance discrepancies compared to the outcomes reported in their original publications. Nevertheless, as the authors have indicated, this comparative analysis is not meant to endorse any algorithm as superior; it is still not yet a complete view of the performance of these models.

**Strengths:**

The strengths of this work include
1)	It provides a valuable discussion of evaluation issues associated with retrosynthesis algorithms, for both single-step and multi-step algorithms. In an era where an increasing number of AI-driven algorithms are being developed for this critical area in chemistry, the establishment of a consistent and meaningful set of metrics  is essential.
2)	The creation of a benchmarking library represents a pivotal initial move toward fostering the creation of more robust and credible AI algorithms for retrosynthesis.

**Weaknesses:**

The weaknesses include
1)	The provided GitHub link is inactive (github.com/anonymous/anonymous), preventing access to SYNTHE-SEUS's operational details and validation of its efficacy in algorithm evaluation.
2)	The SYNTHE-SEUS library falls short in addressing the core evaluation challenges within retrosynthesis. It continues to rely on top-k accuracy without incorporating other significant metrics such as precision and recall, which may offer a more comprehensive evaluation about algorithm performance.
3)	Although the authors purport that SYNTHE-SEUS is intended to serve as a resource for researchers developing retrosynthetic methods, there is a lack of clarity in Section 3 on how the platform will facilitate the flexible incorporation of different feature definitions, such as functional groups and molecular fingerprints.

**Questions:**

The link to doesn’t work: github.com/anonymous/anonymous.  Is it possible to check how SYNTHESEUS works and support the evaluation of different algorithms?

---

> ### Author Response · Authors · 2023-11-10
> **Response**
>
> Thank you for your review! Below we address each of the weaknesses you mentioned one by one. Please let us know if you have further comments or suggestions for what specifically is missing in `syntheseus`.
>
> ---
>
> **(Q1):** The GitHub link doesn't work.
>
> **(A1):** Code is provided in the supplementary material as a zip file. The "github/anonymous/anonymous" link is just a placeholder to maintain anonymity; it will be replaced by the real link after acceptance.
>
> ---
>
> **(Q2):** Only top-k accuracy is incorporated into `syntheseus`, while precision and recall are not.
>
> **(A2):** We discuss in the manuscript that top-k accuracy is equivalent to recall. We also discuss precision as a metric, and suggest that round-trip accuracy using a forward model can be used as a surrogate for it, which `syntheseus` has support for. However, as we say in the paper, precision is not a well-defined metric without a feasibility model, and all such models are imperfect, so the best you can really hope for is to provide a _framework_ that can compute precision with a given model. In `syntheseus` the forward model is abstracted away using a similar interface as the backward model, so the framework is flexible in terms of which model is used.
>
> ---
>
> **(Q3):** Unclear how the platform will facilitate the incorporation of different feature definitions, such as functional groups and molecular fingerprints.
>
> **(A3):** In our library, the interface for a single-step retrosynthesis model is (roughly) a function that accepts a molecule as a SMILES string, and returns a list of suggested reactions to make this molecule, representing the reactants in each reaction again as SMILES strings. This means any kind of model that can parse SMILES strings can be used. For example, `rdkit` could be used to convert the SMILES into a molecule object, from which one can compute fingerprints or extract functional groups.

---

> ### Author Response · Authors · 2023-11-17
>
> Dear Reviewer cznK,
>
> we wanted to ask whether you were able to access our code via the supplementary material, and whether our reply clarifies your questions. Are there any additional changes you would like to see to improve your rating?

---

> > ### Comment · Reviewer_cznK · 2023-11-23
> > **thanks for response**
> >
> > Thanks for the response. We did have a chance to check the code provided in the supplementary zip file.
> > For a benchmarking library, it is essential to conduct a comprehensive evaluation and comparison of the performance of various models. This involves assessing multiple metrics rather than relying solely on one. If experts in the field of chemistry can assist in evaluating the precision or other aspects of result quality, their involvement would be valuable.   Human evaluation has become increasingly prevalent in recent AI studies. Expert assessments of these retrosynthesis results would make a substantial contribution to this research.

---

> > > ### Author Response · Authors · 2023-11-23
> > > **Response**
> > >
> > > We’re happy to hear you were able to take a look at the code. We agree looking at several metrics is important, which is why `syntheseus` supports many metrics (top-k accuracy, MRR, time spent on model inference, round-trip accuracy using a forward model as a proxy for precision).
> > >
> > > While we agree expert evaluation would in principle be useful, it is expensive and difficult to scale, as we discussed in the manuscript (Pitfall S1). We believe that **at an ML conference, lack of expert validation should not be the main reason to reject a paper**, given that the vast majority of accepted papers lack that element. It would also exclude teams of machine learning researchers that do not have chemistry expertise. **It appears that lack of chemist evaluation is your only unresolved concern, and otherwise you think the work is valuable - in that case, would you consider increasing your score?**

---

### Official Review · Reviewer_WK4y · 2023-11-03

**Soundness:** 3 good
**Presentation:** 3 good
**Contribution:** 3 good
**Rating:** 6
**Confidence:** 3

**Summary:**

This paper aims to re-evaluate existing single-step and multi-step retrosynthesis algorithms in a fair basis, with resolving their inconsistent settings and practices. The authors first discussed the evaluation methods adopted in previous works and pointed out their shortcomings, then presented the best meaningful practices. A benchmarking library SYNTHESEUS is then provided to uniform the evaluation practice and re-evaluate existing algorithms.

**Strengths:**

1. The reflection on existing evaluation is important and desired for this community. This paper provides a broad and in-depth discussion on the current progresses, their pitfalls and suggested better practice;
2. The benchmarking library has practical significance for the practitioners to proceed from the same basis, without worrying possible setting inconsistencies;
3. Several existing algorithms are re-evaluated using the standardized protocol to reveal a more faithful comparison.

**Weaknesses:**

1. For existing multi-step planning methods, important related work [1] is not discussed and evaluated. This work also provides a “set-wise exact match” metric, which is related to the discussion of how to evaluate the success of planning. I would like to see the authors’ discussion on this work.
2. While I can see the clear contribution in standardized evaluation, this work could be further strengthened to provide novel metrics or protocols.

[1] Liu, Songtao, et al. "FusionRetro: molecule representation fusion via in-context learning for retrosynthetic planning." International Conference on Machine Learning. PMLR, 2023.

**Questions:**

1. Can the authors discuss the multi-step planning method FusionRetro [1] and its proposed evaluation metric? If it makes sense, can the authors evaluate it and include its metric in the experiment or library?
2. Can the authors provide an overview of the library, e.g., supported single-step models, search algorithms, evaluation metrics etc?

---

> ### Author Response · Authors · 2023-11-15
> **Response**
>
> **(Q1)**: FusionRetro [1] should be discussed and evaluated, and its “set-wise exact match” metric should be included.
>
> **(A1)**: Thank you for the suggestion. We carefully reviewed [1] and found the approach proposed there very interesting and relevant. Thus, we will cite this publication in the discussion section of our paper. We have also reviewed the “set-wise exact match” metric, and drafted utility functions that make computing it easy. We expect to merge the relevant PR into `syntheseus` soon (pending some testing and code review). We will also add support for search algorithms like FusionRetro into our roadmap for the library (however, properly integrating it will take much longer than the length of the rebuttal period).
>
> ---
>
> **(Q2)**: Can the authors provide an overview of the library (supported single-step models, search algorithms, evaluation metrics)?
>
> **(A2)**: The `syntheseus` library (snapshot of which can be accessed through the supplementary material) is, at the top level, split into the following subdirectories:
> - `reaction_prediction`, which defines inference wrappers for all supported single-step model types (7 models mentioned and evaluated in the paper, plus an 8th model currently being integrated as requested by Reviewer JYEf) and a evaluation script `eval.py`, which computes metrics such as: top-k accuracy, MRR, time spent on model inference (excluding e.g. post-processing or model loading, leading to a more precise measurement), round-trip accuracy using a forward model (proxy for precision) and number of model parameters.
> - `search`, which defines all supported multi-step search algorithms (MCTS for both OR and AND-OR graphs, Retro*, and PDVN). The search algorithms are implemented in such a way that the commonalities between them are extracted into common base classes, making it easy to build further algorithms that e.g. reuse the logic for expanding the search graph present in MCTS but modify the node selection criteria.
> - `interface`, which defines data structures at the intersection of single-step and multi-step, making sure they use the same objects for compatibility; notably, `interface/models.py` contains the shared `ReactionModel` interface.
> - `cli`, which defines a top-level search entry point for running end-to-end retrosynthesis.
>
> There are already several `*.md` files in the repository explaining this structure, and a `tutorials` directory with notebooks showcasing how `syntheseus` can be used. On top of this, we are also working on hosting documentation in the form of a static webpage to make it even easier to set up and use the library.

---

> ### Public Comment · ~Songtao_Liu2 · 2023-11-18
> **Thanks a lot for discussing my work.**
>
> Dear Reviewer and Authors,
>
> I am Songtao Liu, the first author of FusionRetro. I noticed that you have been discussing my work, and I have carefully and thoroughly checked this paper (Syntheseus). I believe it is an outstanding work that will undoubtedly inspire future research in this field. I greatly appreciate your discussion of my work.
>
> Best regards,
> Songtao Liu

---

> > ### Author Response · Authors · 2023-11-21
> > **Response**
> >
> > Thank you for your kind comment, we're happy you liked our work!

---

### Official Review · Reviewer_eNX2 · 2023-11-06

**Soundness:** 3 good
**Presentation:** 2 fair
**Contribution:** 2 fair
**Rating:** 5
**Confidence:** 4

**Summary:**

The paper discusses the evaluation of maching learning-based retrosynthesis algorithms. The authors argue that existing evaluation practices have shortcomings and inconsistencies, leading to inaccurate comparisons between methods. To address this, they introduce a benchmarking library called SYNTHESEUS, which enables consistent evaluation of single-step and multi-step retrosynthesis algorithms. The authors use SYNTHESEUS to re-evaluate previous algorithms and the ranking of state-of-the-art models changes. The paper highlights several pitfalls in the evaluation of single-step models and suggests best practices, including measuring precision instead of recall, using consistent and realistic post-processing, reporting inference time, and focusing on prediction with unknown reaction types.

**Strengths:**

+ The paper would contribute to the community. The consistency of evaluating retrosynthesis algorithms is an issue and the paper proposes a fair approach.
+ The paper introduces SYNTHESEUS and re-evaluates state-of-the-art models for retrosynthesis.
+ The paper lists possible pitfalls of previous algorithms including post-processing and measurement, etc.
+ The paper gives the best practices for evaluating single-step models and multi-step models.

**Weaknesses:**

+ The methods included in the re-evaluation are not enough as a benchmarking library. Some common baselines like vanilla LSTM, and vanilla Transformer should be included. Also, more state-of-the-art methods are welcome.
+ It would be better if some case studies were shown to explain and compare corresponding pitfalls and best practices.

**Questions:**

1. What are the criteria for selecting state-of-the-art methods?

2. What are the advantages of Pistachio over USPTO-FULL? It would be better if this issue is discussed in the main
paper.

---

> ### Author Response · Authors · 2023-11-15
> **Response**
>
> **(Q1)**: Some baselines like vanilla LSTM / vanilla Transformer should be included.
>
> **(A1)**: Incorporating baselines in a proper way is significant engineering work, as we do not only run them, but standardize the (conda) environments and dependencies, and wrap using our interface to integrate fully into `syntheseus`. Thus, we are careful to select the baselines to maximize usefulness. The older models are written using legacy frameworks and old tensorflow and pytorch versions, while with RootAligned and Chemformer we have 2 Transformer baselines already, and these papers showed that these models are superior to “vanilla” Transformer models or even LSTMs. Thus, we believe the effort integrating more baselines would be best spent on newer models beating state-of-the-art, rather than old baselines which were already outperformed by many newer approaches. We encourage the community to integrate such models via PRs in our repository if desired.
>
> ---
>
> **(Q2)**: More state-of-the-art methods could be included.
>
> **(A2)**: Since integrating new methods is significant effort, and our work is mostly about providing the evaluation framework, we have to be selective in what we can manage to include ourselves. For the rebuttal, we followed the advice of Reviewer JYEf and integrated the recently published Graph2Edits model (see response to that reviewer for more details). More models will be included after the rebuttal phase, and we invite contributions from the community.
>
> ---
>
> **(Q3)**: There should be case studies to explain and compare the pitfalls and best practices.
>
> **(A3)**: Could you please clarify what you mean by case studies?
>
> ---
>
> **(Q4)**: What are the criteria for selecting state-of-the-art methods?
>
> **(A4)**: We selected the initial set of 7 methods based on a combination of state-of-the-art performance (this is why we leaned towards more advanced Transformer-based models rather than their vanilla counterparts) and coverage of varying model types (e.g. end-to-end, template-based, sequence-of-edits-based). That being said, `syntheseus` is not a “final product” but a library that we will continue to develop (no matter the ICLR review outcome) and open it up to the community, so more methods will be added in the future.
>
> ---
>
> **(Q5)**: What are the advantages of Pistachio over USPTO-FULL?
>
> **(A5)**: There are several advantages of Pistachio:
> - significantly larger than USPTO-FULL and with higher coverage of diverse reaction
> - of high-quality (commercial and carefully curated), and much less noisy
> - continuously updating every few months
>
> In comparison, USPTO is now a bit outdated (1976-2016), and suffers from noise and quality issues caused by automated toolkits (e.g. Indigo) and outdated software that were used to prepare it (see “Recent advances in artificial intelligence for retrosynthesis” for more discussion). Including it would require significant additional preprocessing to give robust results, which would make comparisons to any works using prior versions of USPTO-FULL invalid. That being said, looking at USPTO-FULL can still be worthwhile, and some researchers may not have access to Pistachio. Due to that, we decided it is useful for us to report results on Pistachio as the comparison may be valuable and it is something most researchers cannot compute themselves. Results on USPTO-FULL and other datasets can be easily generated as `syntheseus` now supports evaluation data in several formats, including the format commonly used for USPTO.

---

### Author Response · Authors · 2023-11-22
**Message to all reviewers**

Dear Reviewers,


we would like to thank you again for the time spent reviewing our work. We are near the end of the discussion period, and so far have not heard back from Reviewers eNX2, WK4y and cznK. We would be happy to have further discussion with those reviewers if our rebuttal was not satisfactory to help improve our paper.


To summarise the initial reviews, all reviewers agreed that **our work is crucial for the retrosynthesis community**, acknowledging the **importance of a careful re-evaluation of existing methods**. In general, all reviewers agree that our work is **useful**.


On the negative side, some reviewers asked clarification questions, asking e.g. how the library is structured or how to access the code, which we have addressed. Some reviewers also asked for the inclusion of more single-step models into the benchmark, thus we **integrated Graph2Edits as the 8th model class**. We believe the current selection of model types is solid, and it will keep growing over time, as `syntheseus` is **not a “final product”**, and rather a **library which we plan to continue developing, but to do this most effectively we need to promote it in the community**.


We would kindly ask each individual reviewer to consider whether our response at least partially addressed their concerns, and if so, to consider increasing their score.


Authors

---

### Meta-Review · Area_Chair_YdYr · 2023-12-06

**Metareview:**

This paper introduces a benchmarking library named SYNTHESEUS to re-evaluate existing retrosynthesis algorithms. The reviewers acknowledge the valuable discussions of evaluation issues associated with retrosynthesis algorithms, and the practical significance of the proposed benchmarking library. However, one reviewer points out that the SYNTHE-SEUS library still relies on top-k accuracy without incorporating other significant metrics and the rebuttal can not convince the reviewer. Considering the mixed opinions and identified weaknesses, I lean towards rejection.

**Justification For Why Not Higher Score:**

Please see the meta-review.

**Justification For Why Not Lower Score:**

N/A

---

### Decision · Program_Chairs · 2024-01-16

Reject